# What Can Electrochemical Methods Offer in Determining DNA–Drug Interactions?

**DOI:** 10.3390/molecules26113478

**Published:** 2021-06-07

**Authors:** Sandra Ramotowska, Aleksandra Ciesielska, Mariusz Makowski

**Affiliations:** Department of Bioinorganic Chemistry, Faculty of Chemistry, University of Gdańsk, Wita Stwosza 63, 80-308 Gdańsk, Poland; sandra.ramotowska@ug.edu.pl (S.R.); olaciesielska5@gmail.com (A.C.)

**Keywords:** DNA–drug interactions, drug analysis, electrochemical methods, cyclic voltammetry, differential pulse voltammetry

## Abstract

The interactions of compounds with DNA have been studied since the recognition of the role of nucleic acid in organisms. The design of molecules which specifically interact with DNA sequences allows for the control of the gene expression. Determining the type and strength of such interaction is an indispensable element of pharmaceutical studies. Cognition of the therapeutic action mechanisms is particularly important for designing new drugs. Owing to their sensitivity, simplicity, and low costs, electrochemical methods are increasingly used for this type of research. Compared to other techniques, they require a small number of samples and are characterized by a high reliability. These methods can provide information about the type of interaction and the binding strength, as well as the damage caused by biologically active molecules targeting the cellular DNA. This review paper summarizes the various electrochemical approaches used for the study of the interactions between pharmaceuticals and DNA. The main focus is on the papers from the last decade, with particular attention on the voltammetric techniques. The most preferred experimental approaches, the electrode materials and the new methods of modification are presented. The data on the detection ranges, the binding modes and the binding constant values of pharmaceuticals are summarized. Both the importance of the presented research and the importance of future prospects are discussed.

## 1. Introduction

### 1.1. Interactions between Pharmaceuticals and DNA Chain

Deoxyribonucleic acid (DNA) plays an important role in the functioning of life, as it carries the genetic information of living organisms and some viruses. Since the recognition of the role of nucleic acids in living organisms, the effect of various substances on their structure and function has been studied. Understanding the mechanism of the interaction between the pharmaceuticals and the DNA chain is key to biological research, as it enables the necessary information for pharmaceutical design and development processes to be obtained [1,2,3,4,5,6].

The two DNA strands are linked primarily through hydrogen bonds between complementary nucleobases. Small molecules interact with the DNA helix in several different ways (Figure 1), which primarily include intercalation, major and minor groove interaction, and nonspecific electrostatic interactions with the negatively charged nucleic acid sugar–phosphate structures, as well as covalent bonding [3,7].

Intercalation is a type of noncovalent interaction with DNA involving spatially flat systems sliding in between base pairs in the double nucleic acid helix. The compounds that interact this way generally have flat aromatic or heteroaromatic ring system(s) with a thickness of about 0.2–0.4 nm. As a result of intercalation, the aforesaid systems are positioned perpendicularly to the helical axis. The formed adduct is stabilized by the interactions of the flat aromatic systems with the DNA nitrogenous bases. Intercalators generally do not exhibit base sequence specificity, but are preferably located at sites with a predominance of GC (guanine–cytosine) base pairs instead. After the intercalation process, the primary and secondary DNA structures remain unchanged. Nonetheless, a change occurs in the tertiary structure. Namely, the helix torsion angle bends and the DNA strand becomes stiffened and elongated. An example of a typical intercalator is amsacrine [3,8,9,10]. The combination of two intercalating planar units linked by an alkyl chain gives to more complex bifunctional compounds, called thread-like bis-intercalators (e.g., bis-acridines [11]). In addition, molecules that are composed of three or more such systems have been designed. These compounds are characterized by an increased affinity compared to conventional intercalators, and thus have improved therapeutic properties. Threading intercalation is an unusual mode of DNA binding with significantly lower association and dissociation rates compared to classical intercalation [12,13]. 

Crystallographic studies indicate that under physiological conditions, the DNA double helix is structurally similar to the model form B DNA (10.5 base pairs per turn), the surface of which contains two grooves. The region where the two strands are close to each other is called a minor groove, while the region where the two strands are away from each other is called a major groove. Their dimensions and geometry are therefore important recognition elements for the ligands to bind correctly to DNA. The DNA phosphate–sugar skeleton is flexible, which allows the torsion degree of the double helix to change depending on certain factors. This flexibility is also affected by the number of hydrogen bonds between the complementary bases. The regions rich in GC, the base pairs are more “rigid” compared to the base pairs that are rich in AT (adenine–thymine). A high conformational lability of the ligand structure is required to adjust the compound according to the shape of the groove. A characteristic feature of these compounds is the presence of several single rings connected by a short linker. The stability of the complex formed this way is characterized by the physicochemical interactions (usually hydrogen bonds) between the functional groups of the ligand and the functional groups within the small groove. The compounds interacting with the DNA in the small groove usually have a greater affinity toward the regions rich in AT base pairs. [3,14,15]. Moreover, the molecules that are composed of both polycyclic systems and elastic side chains interact with the DNA helix by intercalation and by binding to the DNA grooves. An example of such compounds is actinomycin D [16].

### 1.2. Techniques Used to Describe DNA–Molecule Interactions

Prior to in vivo research, the interactions between pharmaceutical molecules and DNA can be determined in the chemical laboratory, using the following techniques: spectroscopic methods (NMR [17,18,19], IR [19,20], Raman [21,22], and UV–Vis spectroscopy [23,24,25,26,27,28,29], linear and circular dichroism [19,20,29], spectrofluorimetry [2,25,27,28,29]), mass spectrometry [30,31], equilibrium dialysis [32], surface plasmon resonance (SPR) [33,34], and to some extent, molecular modeling techniques [1,3,6,35,36]. All the above-mentioned methods are generally applicable in the assessment of the position, strength, and mechanism of the interaction, which in turn are crucial to understanding the drug’s mechanism of action. Each of these techniques has a certain range of applicability and information that it can provide. 

The equilibrium dialysis method can be used to measure the amount of ligand bound to a macromolecule [37]. Additionally, the binding isotherms and Scatchard plots used to compare the binding parameters of a drug to nucleosomes and DNA can be estimated from such an experiment as well [32]. However, equilibrium dialysis is an indirect method and requires the support of other techniques to describe the interactions in a comprehensive manner. SPR is an optical technique that allows the concentration of biomolecules to be determined by measuring the changes in the light refraction parameters. The basis of the SPR is the interaction of an incoming light source with a thin metallic film in close contact with a prism or grating. In order to detect an interaction, one molecule is immobilized onto the sensor surface and its binding partner is injected in a sample buffer [37,38,39]. The main advantages of SPR over other methods is that no labeling is required, the amount of both ligand and analyte needed to obtain satisfying results is low, and the experiment is relatively rapid. On the other hand, the limitation of this technique is that it cannot verify the stability of the complex formed during drug binding to DNA [34]. Structural analysis tools coupled with molecular modelling techniques have had a considerable impact on the understanding of the microscopic structural heterogeneity of DNA and constitute a basis for compound-DNA recognition. However, these techniques are primarily used to study the adduct structure rather than determining the bond constant values [19,31,36]. Spectrophotometric techniques are useful due to their low sample consumption and their ability to provide information regarding the binding affinity. Changes in the spectrum of the studied drug in the presence of different DNA concentrations allow the DNA–drug binding mode and the value of binding constant [25] to be determined. However, they cannot be used for compounds which do not have absorption maxima in the tested range or their absorption maximum coincides with the maximum absorption of DNA. If the properties of the studied compound allow it, both spectroscopic and electrochemical methods are used to describe the interactions as fully as methodologically possible [24]. Electrochemical techniques are extremely advantageous in the case of compounds that cannot been studied and described by either UV–Vis or fluorescence spectroscopy because of limitations such as the weak intensity of absorption/fluorescence maxima or the overlap of the electronic transition bands of the studied compound with the electronic transition bands of the DNA. 

The recent increase in the use of electrochemical methods is due to their numerous advantages, the most important being a high sensitivity, selectivity in relation to electroactive molecules and a wide range of linearity. On the other hand, ergonomic advantages are related to the relatively low costs of both the measuring equipment and the single analysis itself, the low harmfulness of the reagents used in the analysis and the short time needed to perform the experiment. Compared to the other methods, these techniques require only a small amount of the sample and are characterized by a high reliability [1]. The apparatus used in this technique is portable and the electrodes can be modified to increase the sensitivity and can be miniaturized for mass production. Changes in the electrochemical parameters registered with the use of voltammetric techniques provide a large amount of information about the studied process. The binding of the drug to a cellular target can be converted into a useful electrical signal, electron transfer, and a potential or impedance change at the electrode–solution interface. Due to the similarity between the electrochemical and the biological redox processes, the oxidation mechanisms occurring at the electrode and in the body may share similar principles [40]. 

## 2. Electrochemical Approach to DNA–Drug Interaction Description

### 2.1. Principles of Measurement with Electrochemical Techniques

The binding of a pharmaceutical compound with DNA is most often observed by differences in the redox process (electrochemical behavior) of a given drug in the absence and presence of DNA. Changes mainly include the shifts in the formal potential of the redox couple and the decrease in the peak current, resulting from the sudden decrease in the diffusion coefficient after binding to DNA [1]. The voltammetric methods involving the electrolysis of the diffusion layer and the current measurements (I [A]) versus the applied electrode potential (E [V]) are widely used.

To observe and interpret the signals from the DNA–molecule adduct, it is important to understand the electrochemical behavior of both the studied compound and the DNA. All DNA bases are oxidized at the glassy carbon electrode (GCE). Guanine and adenine bases are oxidized at much lower positive potentials compared to cytosine and thymine. For example, in the Satana et al. [41] experiment, the double-strand DNA (dsDNA) exhibited two well-defined peaks in an acetate buffer with a pH of 4.5, which were observed by differential pulse voltammetry (DPV). The peaks corresponded to the oxidation reactions of deoxyguanosine (dGuo) and deoxyadenosine (dAdo), at the potential values (E_pa_) of +0.98 V and +1.24 V, respectively. The oxidation of adenine and guanine is a two-step process associated with the loss of four electrons and four protons.

### 2.2. Types of Electrodes and Research Approaches

In voltammetric measurements, a three-electrode measuring system consisting of a working electrode, a reference electrode (Ag/AgCl or saturated calomel electrode), and an auxiliary electrode (Pt wire) is used. The stationary working electrode is composed of various types of electrode materials. The most common are glassy carbon electrodes (GCEs) [42,43,44,45,46,47,48,49], pencil graphite electrodes (PGEs) [50,51,52,53,54,55,56], carbon paste electrodes (CPEs) [57,58], hanging mercury drop electrodes (HMDEs) [59], platinum electrodes [60], and gold electrodes [61,62]. These electrodes can be used either in the bare form [44,54,63] or modified with a layer capable of increasing their sensitivity [50,51,52,53,55,56]. 

In addition, the approach to conducting the experiment can vary depending on the electrochemical techniques used. Such analysis often involves examining the electrochemical behavior of a pharmaceutical in a solution to which DNA is gradually added. Consequently, the impact of the subsequent portions (increasing the DNA concentration) on the redox processes of the examined system is determined [24,26,44,46,59,63]. An alternative method is the modification of the electrode, which involves immobilizing one of the analyzed system elements (studied compound with biological activity [64] or DNA [41,42,57,61,62,63,65,66]) on the surface of the working electrode. The last-mentioned approach is often used to obtain DNA biosensor systems for monitoring drug interactions.

### 2.3. Electrochemical Biosensors

A chemical sensor is a device that transforms chemical information into an analytically useful signal. The recognition system utilises a biochemical mechanism in biosensors [67]. They usually contain two basic components: a molecular recognition system (receptor) and a physicochemical transducer [68], or an electrochemical transducer in the case of an electrochemical biosensor that can be considered a chemically modified electrode [69]. This integrated receptor-transducer device is able to provide selective and quantitative analytical information using a biological recognition element. An electrochemical DNA-based biosensor in turn, is a device that integrates nucleic acids as the biological recognition element and integrates an electrode as the physicochemical transducer [70,71,72,73]. 

There are several types of natural and synthetic DNA and RNA molecules available for electrochemical biosensors, including chromosomal DNA as well as well-defined viral or plasmid nucleic acids. Their preparation method is of great practical importance. This allows the sensor to be prepared for the appropriate application. Electrochemical biosensors are also widely used outside the analysis of DNA–molecule interactions in the detection and quantification of chemicals such as drugs, metabolites, pollutants, biomarkers etc [69,74,75,76,77,78].

## 3. Electrochemical Methods Applied

### 3.1. Cyclic Voltammetry (CV)

Cyclic voltammetry (CV) is widely used for determining the interactions of small biologically active molecules with DNA [2,3,24,26,42,59,61,79,80,81,82]. During the measurement, a linearly changing potential is applied to the working electrode. When the target value of the potential is reached, a change in the electrode polarization direction takes place. This allows the reversibility of the analyzed redox process to be observed. The potentiostat enables the precise polarization of the working electrode and the measurement of the current flowing between it and the reference electrode. The shape of the resulting voltammogram is influenced by the type of redox system studied and the conditions of the measurement. The primary factors determining the voltammogram form are the speed of the depolarizer molecules transported to the electrode surface and the speed at which the electrode functions (kinetics of the oxidation/reduction process). In fact, the electrode does not function until the applied potential reaches the specific value at which the redox reaction occurs, which is manifested by an increase in the current. The driving force of this process is diffusion, which is caused by a gradient of the analyte concentration in the electrode layer and in the further part of the solution. Over time, the ions from further layers of the solution must reach the electrode; in other words, the number of depolarizer particles reaching the electrode decreases. Thus, the current decreases after reaching the maximum value. Subsequently, the recorded signal reaches its peak, which is indicated by the values of “peak potential” (Ep) and “peak current” (Ip) [83,84]. CV is often used to describe the chemical characteristics of compounds [85]. It provides information, for example, on the effect of protonation and the formation of hydrogen bonds on redox processes [86]. It also determines the acid dissociation constants [87] and acts as a tool for analyzing the complexation process [88]. 

CV is one of the voltammetric techniques which allows for the prediction and evaluation of the interactions, the binding strength, and the DNA damage caused by biologically active molecules targeting the cellular DNA. The most important information provided by these methods are: (i) the diffusion coefficient values of both the molecule and its adduct with DNA; (ii) the binding affinity, which is expressed as the value of the binding constant (*K*); (iii) the type of interaction mode; and (iv) the size of the binding site (s) where the drug–DNA interactions occur. 

The diffusion coefficient of the electroactive species can be determined using the dependencies defined by the Randles–Ševčík equation (for 25 °C temp.): (1)Ip=2.69×105n32·A·C·D12·v12
where *n* is the number of electrons transferred during the redox process, *A* is the electrode area in cm^2^, *D* is the diffusion coefficient in cm^2^ s^−1^, *C* is the concentration in mol·cm^−3^, and ν is the scan rate in V/s. Moreover, the linear plots of $I_{p}$ vs. ν^1/2^ provide evidence for a diffusion-limited mechanism of the redox process. The value of the binding constant can be determined from the following equation:(2)$$log\;\left(1/\left[DNA\right]\right)=logK+log\left(I/I_{0}-I\right)$$
where *K* is the binding constant, and *I*_0_ and *I* are the peak currents of the studied redox process in the absence and presence of DNA, respectively. The binding constant is easily calculated from the intercept of the plot of log (1/[DNA]) versus log (*I*/(*I*_0_−*I*)). A high *K* value suggests intercalation-based interactions, whereas a low value implies a rather weaker groove or electrostatic interactions. The type of interaction mode can also be determined by the potential shift direction. The binding site size indicates the number of free base pairs in dsDNA interacting with the studied compound. This can be determined by performing a linear regression analysis of the experimental data according to the following equations:(3)$$\frac{C_{b}}{C_{f}}=\frac{K\;\left[DNA\right]}{2s}$$
(4)$$\frac{C_{b}}{C_{f}}=\frac{I-I_{DNA}}{I_{DNA}}$$
where *C_b_*/*C_f_* is the concentration ratio of the bound and free compounds and s is the size of the binding site.

The essential parameters of all the research on the electrochemical aspects of the DNA–drug interactions described in the paper are summarized in Table 1. It presents the applied electrochemical methods, the electrode materials, and the media in which the discussed studies were conducted. Moreover, it contains the values of the limits of detection (LOD) and/or the limits of quantitation (LOQ), and the binding constant, as well as information about a suggested interaction mechanism.

In their studies on 1,5-di(piperazin-1-yl)anthracene-9,10-dione (1,5-ppz-AQ), Białobrzeska et al. [24] determined the intercalation mechanism by both spectroscopic and electrochemical methods. The intensity of the studied anthraquinone derivative cathode peak highly decreased (from *I* = 10 μA to *I* = 0.4 μA) with an increased DNA concentration (up to 100 μM) (Figure 2). Moreover, the lower values of the diffusion coefficient determined for 1,5-ppz-AQ in the presence of *calf thymus* DNA (ctDNA) confirmed the formation of an adduct. The value of the binding constant was calculated from the intercept of the plot of log (1/[DNA]) versus log (*I*/(*I*_0_−*I*)) (Figure 3). The linear fitting of the voltammetric data yielded the *K* values 1.94 × 10^5^ M^−1^ and 1.96 × 10^5^ M^−1^ for 1,5-ppz-AQ and ethidium bromide (measured by the same method for comparison), respectively. These values indicated that the compounds had a high affinity to the DNA. Furthermore, the linear fit analysis yielded a site size *s* of 1.08. A higher binding constant value of 1.5-ppz-AQ, compared to the values of the various intercalating drugs that are clinically used (epirubicin, mitoxantrone, etc.), suggests its potential use as an anticancer drug.

The lower values found for the binding constant (*K*) between the studied compound and the DNA chain indicate the groove bonding interaction mode. Qin et al. [89] determined the binding constants of two 5-fluorouracil derivatives that interacted with nucleic acid (*K* = 2.33 × 10^3^ M^−1^ and *K* = 1.45 × 10^3^ Mp^−1^ for *ortho-* and *meta*-substituted derivatives, respectively). Moreover, molecular docking was utilized to simulate the modes of the interactions between the drugs and the DNA. The obtained results demonstrated that the studied compounds acted as groove binders and interacted with the nucleic acid chain by binding to the minor groove of the DNA double helix.

The value of the binding constant is not the only factor used to assess the type of interactions. In addition, the direction of the signal shift on the voltammogram, with an increase in the DNA concentration, indicates the nature of the interactions [91]. In general, the positive shift (a shift toward higher potential values) is caused by the intercalation with DNA [92], while the negative shift is observed for the electrostatic interaction of a drug with DNA [93]. Based on these trends, for example, the interactions of valrubicin with the DNA chain were determined [94]. Upon the successive addition of DNA to the valrubicin solution, the redox peak currents decreased and shifted to positive values due to the formation of a DNA–valrubicin adduct with a smaller diffusion coefficient. Therefore, the anodic shift in the voltammetric characteristic of valrubicin and the decrease in the peak currents with the addition of DNA were attributed to the intercalation mechanism. 

Bayraktepe [54] used CV to study the interaction of DNA with dasatinib (DSB), an anti-cancer drug, for chronic myelogenous leukemia and acute lymphoblastic leukemia treatment. She registered CV voltammograms of DSB in an acetate buffer solution (pH 4.8) using a pencil graphite electrode. A DSB anodic peak occurred at the +0.90 V potential and it corresponded to the two-electron oxidation of the sulfur group of the thiazole ring to sulphonyl. The logarithm of the peak current vs. the logarithm of the scan rate plot had a linear character (R^2^ = 0.9923) which indicated that this redox process was controlled byp adsorption under diffusion conditions. The DSB peak intensity decreased and shifted toward positive potential values in the presence of dsDNA. The calculated values of the diffusion coefficients were 8.40 × 10^−5^ cm^2^ s^−1^ and 4.59 × 10^−5^ cm^2^ s^−1^ for the free DSB and the DSB–DNA adduct, respectively, which indicated the drug–DNA binding. Moreover, the heterogeneous electron transfer rate constants (k_s_) and the electrode surface concentration for the DSB and the DSB–DNA complex were calculated.

In addition, the interaction of metal–ligand complexes with the DNA chain was also studied using the electrochemical methods. Jabeen et al. [82] analyzed three flavonoids (Fls), namely morin (mor), quercetin (quer), and primuletin (prim), as well as their complexes (with Cu (II) and Fe (III)) and investigated their DNA-binding ability. All the complexes were designed and tested for anticancer potential relative to flavonoid ligands. The values of the binding constant (*K*) and the DNA binding modes were determined through spectroscopic methods and CV. The results of the conducted experiments showed that the complex compounds exhibited different binding modes compared to the corresponding flavonoids. These differences, in turn, strongly influenced the apoptotic activity of flavonoids as well as their metal complexes. It was found that Fe–mor, Cu–quer, prim, and Fe–quer were bound to dsDNA through the electrostatic mode of binding, while Cu–mor, mor, Cu–prim, and Fe–prim intercalated into it, whereas quercetin was shown to interact with the DNA groove.

### 3.2. Differential Pulse Voltammetry (DPV)

In voltammetric techniques, the measured electrochemical signal is an algebraic sum of the undesirable capacitive current and the desired current related to the proper electrode reaction, the so-called Faraday current. DPV is an electrochemical technique which is widely used in chemical analysis and for studying the interactions of pharmaceuticals with DNA [41,43,45,51,60,63,65,94,95]. Its high sensitivity results from the pulsating change of potential applied to the working electrode. This type of signal modification effectively eliminates the capacitive current, thus facilitating the analysis of substances in a lower concentration range compared to the CV technique. The shape of the voltammetric curve is determined by a series of potential pulses applied at the right time to the working electrode. Following each pulse, the potential value returns to a slightly more negative value in the cathode part and to a more positive value in the anode part compared to that before the pulse. The pulse techniques work on the principle that with the step-change in potential, the values of both the currents increase sharply, while decreasing at different speeds. The capacitive current decreases rapidly compared to the Faraday current [84].

Buoro et al. used DPV for the electrochemical study of the interaction between gemcitabine (GEM) and DNA [43]. No GEM-associated redox process was observed under the experimental conditions. Two different approaches were used for studying the interactions: an unmodified GCE and a DNA electrochemical biosensor, prepared by successively covering the GCE surface with drops of the dsDNA solution. The DP voltammogram recorded immediately after the addition of GEM to the dsDNA solution displayed a decrease in the oxidation peak currents of dGuo and dAdo, compared with the control dsDNA solution. This effect was enhanced with an increase in the duration of the incubation of the sample and occurred under both experimental conditions (unmodified electrode and DNA electrochemical biosensor). The changes resulted from the aggregation of dsDNA, caused by the interaction with GEM, and were consistent with the spectrophotometric measurements. The formation of rigid DNA–GEM structures hindered the nucleoside residues from interacting and oxidizing at the GCE surface. The authors reported that the interaction between DNA and GEM caused modifications in the morphological structure of DNA. The mechanism of the DNA–GEM interaction occurred in two successive stages. The first stage was independent of the DNA sequence and led to the aggregation of dsDNA and the formation of the GEM–DNA rigid structure. The second stage favored the interaction between guanine hydrogen atoms in the CG base pair and fluorine atoms on the GEM ribose moiety, which induced the release of guanine residues on the electrode surface.

A similar approach was used by Diculescu et al. [45] in an experiment for analyzing the interaction between the anticancer drug danusertib and DNA. The studied drug was itself electrochemically active, which enabled the tracking of its individual signal changes. In addition, the experiment was carried out in incubated solutions, and DP voltammograms were recorded after different incubation periods. The voltammograms recorded after adding danusertib to the dsDNA solution displayed two oxidation peaks (D1 and D2) that were characteristic of the drug at lower potential values compared to the subsequent oxidation peaks of dGuo and dAdo. With a prolonged incubation period, a decrease in the peak current of the D2 signal was observed, while the intensity of peak D1 remained unchanged. An increase was observed in the intensity of the deoxyribonucleosides signals, which was in agreement with the conformational modification of the dsDNA. The second approach to the experiment involved the use of a prepared DNA electrochemical biosensor which had dsDNA immobilized on the GCE surface. The recorded voltammograms demonstrated the formation of a DNA–danusertib adduct (Figure 4A). The effect of drug concentration was also studied (Figure 4B). The binding of danusertib led to modifications in the morphological conformation of dsDNA, causing slight changes in the oxidation peak currents of dGuo and dAdo. 

The studies determined the interaction of dsDNA–danusertib that occurred in two successive stages. The first stage involved the electrostatic interaction of the positively charged piperazine ring with the DNA phosphate backbone. In the second step, the formation of a DNA–drug complex involving the pyranopyrazole moiety occurred, resulting in morphological modifications in the DNA double helix.

The changes in the current signals recorded using the DPV technique can also be used to calculate the value of the drug–DNA binding constant. Dindar et al. [47] studied Citalopram (CIT) and its S-enantiomer—escitalopram (ESC), which are antidepressants belonging to the selective serotonin reuptake inhibitors class. The experiment was conducted by adding increasing concentrations of drugs (from 2 to 10 µg/mL) to the 100 µg/mL ctDNA in an acetate buffer solution with a pH of 4.7 and recording the oxidation signals of dGuo and dAdo using GCE. Based on the reduction in the intensity of the current response (I) caused by the binding of DNA to the CIT and ESC molecules, the plots of log $\left[\frac{I_{complex}}{I_{DNA}-I_{complex}}\right]$ vs. log C_drug_ were determined. Based on the slope and the intercept of the plot values, the binding constants for CIT and ESC were calculated (K_CIT-DNA_ = 5.6 × 10^4^ M^−1^ and K_ESC-DNA_ = 8.5 × 10^4^ M^−1^), using the following equation:(5)$$\left[\frac{I_{complex}}{I_{DNA}-I_{complex}}\right]=-\mathrm{nlogK}\;-{\;\mathrm{nlogC}}_{\mathrm{drug}}\;.\;$$
where I_DNA_ and I_complex_ are dAdo are the peak currents in the absence and presence of different drug concentrations, respectively. Slightly lower values of the binding constants compared to typical intercalators suggest a groove or an electrostatic binding mode rather than an interaction; however, it did not exclude it.

Bayraktepe [54] used DPV to describe the interaction of DNA with dasatinib (DSB) and to determine the adduct binding constant value. In her experiment, a 10.0 μM DSB solution and an acetate buffer solution with a pH of 4.8 was used, and dsDNA was added (from 2 to 70 μM). DPV voltammograms showed that the peak current of DSB decreased with increasing DNA concentrations up to 30.0 μM and then remained constant (Figure 5). Moreover, the peak potential of DPV voltammograms changed to more positive values. The binding constant of the DSB–DNA complex was calculated as K = 2.51 × 10^4^ M^−1^. Moreover, the Gibbs free energy (ΔG°) of the adduct was estimated as −25.10 kJ/mol, using the following equation:(6)ΔG°=−RTInK

The negative value proves the DSB and DNA interaction and indicates that binding occurred spontaneously. All the obtained results indicate that DSB interactions with DNA may have an intercalation mode. Thermodynamic parameters found from voltammetric measurements are comparable to those obtained by the UV spectroscopic method.

Ponkarpagam et al. [49] studied the interactions between ctDNA and rosiglitazone (RG)—a thiazolidinedione anti-diabetic drug—in a 0.05 M Tris-HCl buffer solution (pH 7.3) in the absence and presence of increasing concentrations of ctDNA, using GCE. The decrease in the peak current suggested an interaction of RG with ctDNA by the forming of an electrochemically non-active adduct. This is due to the low diffusion coefficient resulting from the Stokes-Einstein equation and, consequently, from low or negligible currents. The shift of the peak to a more negative potential indicated the groove binding mode of the interaction, which was also confirmed by molecular docking. The binding constant has been determined as K = 3.4 × 10^3^ M^−1^.

Due to their ease of use, the construction of disposable measurement systems is an interesting trend in electrochemical approaches. In particular, modified surface electrodes are designed to improve the sensitivity or selectivity of measurements. Single-use modified biosensors are sensitive, time-saving, and practical tools for detecting the analyte. Such systems have several main advantages: a large surface area, effective mass transport, controllability, and their ability to study interactions in solution. Eksin et al. [51] studied the interaction between daunorubicin (DNR) and ctDNA at the surface of disposable carbon quantum dot-modified PGEs (cQD-PGE). For monitoring the surface-confined interaction, ctDNA was first immobilized onto the electrode surface, and then the electrochemical detection of the interaction between DNA and DNR was carried out. The study aimed to optimize the experimental conditions, such as the concentrations of both ctDNA and DNR, as well as determine the effect of interaction time (from 3 to 15 min) on the changes in the oxidation signals of guanine and DNR. Under optimal conditions, very low values of the detection limits were obtained for DNR and ctDNA—0.02 μg/mL and 0.89 μg/mL, respectively.

Other examples of interesting modifications were presented by Findik et al., who have modified pencil graphite electrodes (NFs-PGE; Figure 6) as sensitive electrochemical biosensors for the anticancer drugs daunorubicin (DNR) [53] and mitomycin C (MC) [52]. In these studies, newly designed and different organic–inorganic hybrid nanoflowers were used.

In the case of the development of disposable voltammetric sensors for the electrochemical analysis of ctDNA, DNR, and the interaction between them, the L-glutamic acid nanoflowers (ga-NFs) and L-cysteine nanoflowers (c-NFs) were applied. Amino acid-Cu_3_(PO_4_)_2_ hybrid NFs were modified at the surface of single use PGE. The c-NFs-PGE electrode turned out to be very sensitive for the detection of both DNA (0.93 µg/mL) and DNR (2.93 µM). In the case of the DNR–DNA interaction, which was the main purpose of Findik’s study, it was determined that both the DNR oxidation peak and the guanine peak decreased at all interaction times. The highest decrease in a short time of 1 min showed that c-NFs-PGE is a very useful sensor for DNR studies. 

The sensors developed to determine MC and its interactions with DNA used glycine and lysine nanoflowers, and were labeled as GNFs and LNFs, respectively. Nanoflowers formed the mono-dispersed 3D hierarchical superstructures (Figure 7). The average diameter of these hybrid NFs with excellent monodispersity was determined to be 3 μm and they were obtained in a homogeneous structure.

The detection limit of the biosensor was determined (1.09 μg/mL for ctDNA) and the biointeraction between MC and ctDNA was investigated.

In order to develop a sensitive tool for DNA detection and to elucidate its structural changes after the interaction with drugs, Bolat [55] constructed a DNA biosensor based on electrodeposited cetyl trimethylammonium bromide-multiwalled carbon nanotubes (poly(CTAB-MWCNTs)) composite on single-use PGE. The DPV and UV–Vis techniques were used to study the interaction of dsDNA with the anticancer drug irinotecan (CPT-11). Voltammetric measurements were based on the changes at the guanine oxidation peak. A high sensitivity was obtained for DNA and DNA–anticancer drug interaction with detection limits of 3.06 μg/mL and 1.03 μg/mL, respectively. Moreover, the binding constant value was determined as K = 6.84 × 10^4^ M^−1^. The experiment showed that the interaction between CPT-11 and DNA leads to a condensation of the DNA double helix and indicated a groove binding mechanism. 

Janiszek et al. [48] in their experiment compared two prospective anticancer drugs, 6-(1*H*-imidazo [4,5-b]phenasine-2-yl)benzene-1,3-diol (IPBD) and its -Cl derivative (Cl-IPBD) with doxorubicin, a widely used anthracycline anticancer agent. For the comparison of the DNA interactions with the drugs, plasmid modified GCEs were used. The aim of the modification of the electrode with supercoiled plasmid instead of typically chromosomal DNA was to minimize the interference of the DNA oxidation. Plasmid (scpUC19) accumulation resulted in the formation of well defined, reproducible plasmid DNA layers on a typical, easily available GCE. In this experiment DPV, square wave voltammetry (SWV) and the less frequently used alternating current voltammetry (ACV) with phase detection 0°, ACV (0°), as well as 90°, ACV (90°) techniques were used in a specific combination. The correlation of the redox signals of IPBD and Cl-IPBD, with their biological effect on cancer cells were shown. Moreover, the effect of Vitamin C on the redox signals of Cl-IPBD that resemble the reduction in Pt(IV) anticancer prodrugs to Pt(II) compounds was observed.

Moreover, biopolymers are used for electrode modification as they offer stable, biocompatible, and large surface areas for the immobilization of biomolecules. Congur et al. [56] modified PGEs with Levan (LVN), a fructan homopolysaccharide comprised of β-d-fructofuranose residues linked by β-(2→6) glycosidic bonds (Figure 8). The aim of the experiment was to develop disposable electrochemical biosensors for the detection of DNA, daunorubicin (DNR), and the biomolecular interaction of DNR with DNA. The interaction of 20 μM DNR with DNA at the DNA-LVN-PGE modified electrode was evaluated between 3 and 10 min and a decrease in guanine and DNR signals (increasing with the interaction time) was observed. This was caused due to the intercalation of DNR into double stranded DNA resulting in strand breaks.

Javar et al. [58] developed an electrochemical DNA biosensor based on modified CPEs (Eu^3+^-doped NiO/CPEs) for the determination of the anti-cancer drug amsacrine. The powder XRD technique was used to examine the crystal structure of the synthesized nanocomposite and cyclic voltammograms of Fe[CN]_6_^3-/4-^ redox couple were recorded at the surface of the bare CPE. NiO NPs/CPE, Eu^3+^-doped NiO/CPE, and dsDNA/Eu^3+^-doped NiO/CPE were used as the indicators for modification. The effect of the amsacrine–guanine interaction has been electrochemically investigated in comparison to the alterations in the guanine oxidation peak in the absence and presence of amsacrine.

The DPV technique can also be used to study the interaction between metal–ligand complexes and the DNA chain. In the experiment carried out by Kumar et al. [60], the results of voltammetric and spectroscopic studies confirmed that tetraazamacrocyclic complexes interacted with DNA through the same type of binding. Voltammograms obtained for each macrocyclic complex displayed a significant decrease in the current intensity in the presence of ctDNA, which indicates that these metal ions stabilize the ctDNA duplex by the intercalation mode. It was found that the macrocyclic cobalt (II) ion complex interacts most strongly with ctDNA.

### 3.3. Other Methods

Other electrochemical methods, including square wave voltammetry (SWV) [59] and electrochemical impedance spectroscopy (EIS) [51,90], are also used to monitor the binding of molecules to the DNA chain. The first one is a type of linear potential sweep voltammetry involving a combined square wave and a staircase potential applied to a stationary electrode. Using SWV, Temerk and Ibrahim [59] determined the effect of a constant dsDNA concentration on the electrochemical response of a series of flutamide (Flu) solutions with varying concentrations. The plot displayed a slope representing the dependence between log [Δ*I/(*Δ*I_max_−*Δ*I*)] and log [Flu], which confirmed the formation of a complex with 1:1 stoichiometry. In addition, the molar relation of flutamide interacting with a molar quantity of dsDNA bases and the binding constant for the dsDNA–Flu adduct were determined. 

EIS involves the measurement of the impedance between the working electrode and the auxiliary electrode, and is used, for example, to assess the structure of a modified electrode surface. Bolat [55] recorded the Nyquist diagrams of the impedance spectra during poly(CTAB-MWCNTs) modification of PGE (Figure 9B). The unmodified PGE electron charge transfer resistance (Rct) value was estimated as 1131.5 Ω and decreased to 15.7 Ω for poly(CTAB-MWCNTs)/PGE confirming the good electrical conductivity of the produced film. This demonstrated the stronger electron transfer ability of the redox ions to the electrode surface. At the next modification step—dsDNA immobilization, an increase in the Rct value was observed (132.2 Ω). This indicates a reduced ability for electron transfer at the electrode surface due to the non-conductive dsDNA layer. In addition, the subsequent modification steps were also controlled using the CV technique (Figure 9A). The registered changes in the CV voltammograms and the Nyquist diagrams of the studied electrode illustrate and confirm the ongoing modification increasing the sensitivity of PGE. 

Eksin et al. [51] used EIS to investigate the surface-confined interaction of DNR with ctDNA. Nyquist diagrams were recorded before and after the interaction process at different intervals. In the Tajik et al. [90] experiment, taxol interacted with guanine and adenine at the surface of the dsDNA-modified PGE. The electrochemical characterization of bare PGE, dsDNA/PGE, and taxol–dsDNA/PGE was made with EIS to illustrate the changes caused by the intercalation-mode binding of taxol to DNA on the electrode surface. The conducted research led to the design and development of a novel taxol biosensor. Moreover, Findik et al. [52,53] used this method to characterize the surface and evaluate the effectiveness of PGE electrode modification with newly designed hybrid nanoflowers. 

A new, unique, and innovative method for studying the interaction of macromolecules is based on a dynamic electrical switching of the DNA layers on a metal electrode surface during solution flow. This technique applies regenerable chips adapted to functionalization. Electrically switchable nanolever technology (switchSENSE) has been used successfully in research, e.g., the binding and dissociation kinetics parameters of molecules such as proteins or polyamides to nucleic acids [96,97,98,99]. The interactions of small molecules with proteins attached to DNA fragments (on the electrode surface) are also being studied using this technique [100]. In our laboratory, we have recently started working on adapting this technique to study the mechanism of binding small drug molecules and their complexes directly to the DNA chain. Initial research results are promising, and the technique has the potential to become another powerful electrochemical-based tool for studying the affinity of pharmaceuticals to biomolecules.

## 4. Conclusions

Research results compiled in this review show that electrochemical methods are a powerful tool in DNA–drug interaction studies. The essential parameters of the experiments are summarized in Table 1. The data presented there illustrate the research directions and possibilities offered by voltammetric techniques.

The selection of the appropriate voltammetric technique and electrode material, the properties of the studied drug, and the specificity of the experiment performed present an opportunity to obtain a distinct set of information on the type and strength of the interaction. This summary demonstrates that CV and DPV techniques are the most frequently used for DNA–drug interaction studies. These methods allow the experiment in selected environmental conditions (buffer type and pH) to be conducted and important parameters such as: limits of drug detection, values of diffusion coefficients, and binding constants to be defined. The thermodynamic parameters including Gibbs free energy and enthalpy changes can be calculated as well [54]. The most common working electrodes used in the study of DNA–drug interactions are often made of carbon materials (GCEs and PGEs). In order to increase the sensitivity of the measurements, various modifications of the electrode surface are made. The most common way is the immobilization of DNA fragments, usually in a double-stranded form (dsDNA), on the working surface [45,90]. The compiled data show that due to their properties and disposability, PGEs are most willingly modified. The introduction of polymeric systems on the electrode surface is just one example of such modifications [56]. Because of the unique chemical and electronic properties of nanomaterials, the detection of biological compounds on nanomaterial-modified electrodes has received especially great attention [51,52,53,55]. Their usefulness is related to their large surface area, high conductivity, and their ability to promote electron transfer rate and stability. The modification of working electrodes extends the preparation time but can significantly increase their usability. In the case of the cQD-PGEs experiment [51], lower LODs values were obtained than in earlier reports related to daurorubicin detection. All the presented experiments were conducted in the pH range from 4.5 to 7.4, which is also suitable for in vivo studies as it is compatible with intracellular conditions. To control the course of modification and to analyze the surface of the modified electrode, EIS and SVW techniques measurements are preferred. Most of the conducted studies concern anticancer drugs. 

Understanding the mechanism of DNA–drug interactions is crucial in biological studies on drug design and pharmaceutical development processes. The use of voltammetric techniques is extremely useful for this purpose. They provide the ability to study the interactions of potential drugs with DNA in a comprehensive manner. Electrochemical biosensors are both sensitive and convenient in application. They can be used for many different pharmaceuticals, especially if they contain an electrochemically active moiety. Voltammetric methods are very important as both primary and complementary analysis tools in in vivo studies, aiding in the exploration of the nature of DNA–drug bonding. Due to their usefulness, they are expected to become even more popular in the future. New adjustments to biosensors will most likely include modifications to biosensors and the creation of regenerable chips that can be adapted to any desired functionalization.

## Figures and Tables

**Figure 1 molecules-26-03478-f001:**
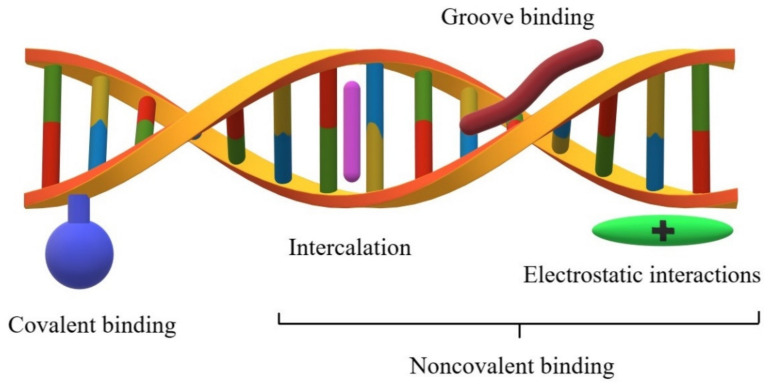
Types of interactions between molecules and the DNA chain.

**Figure 2 molecules-26-03478-f002:**
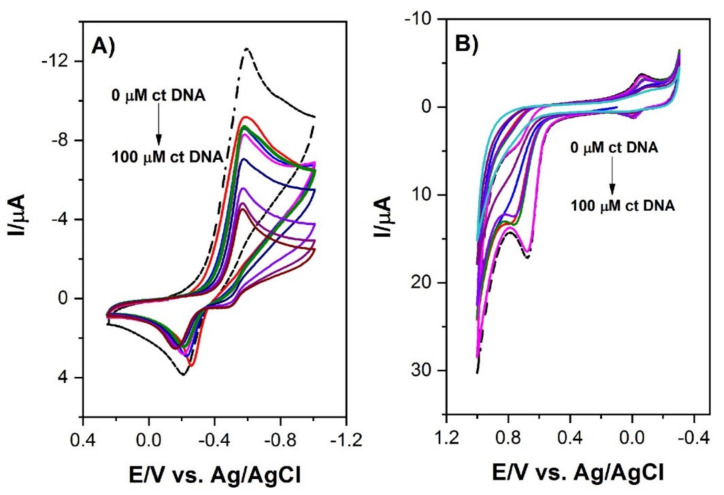
(**A**) Cyclic voltammogram of 2 × 10−4 M 1,5-ppz-AQ in an aqueous buffer at pH 7.4 in the absence (dashed line) and the presence (solid line) of 10–100 μM ct-DNA on the glassy carbon electrode. Scan rate: 100 mV s−1, temperature: 25 °C; (**B**) Cyclic voltammogram of 2 × 10−4 M ethidium bromide in aqueous buffer at pH 7.4 in the absence (dashed line) and the presence (solid line) of 10–100 μM ctDNA on the glassy carbon electrode. Scan rate: 100 mV s−1, temperature: 25 °C. Figure adapted from the reference [24] with permission from Elsevier.

**Figure 3 molecules-26-03478-f003:**
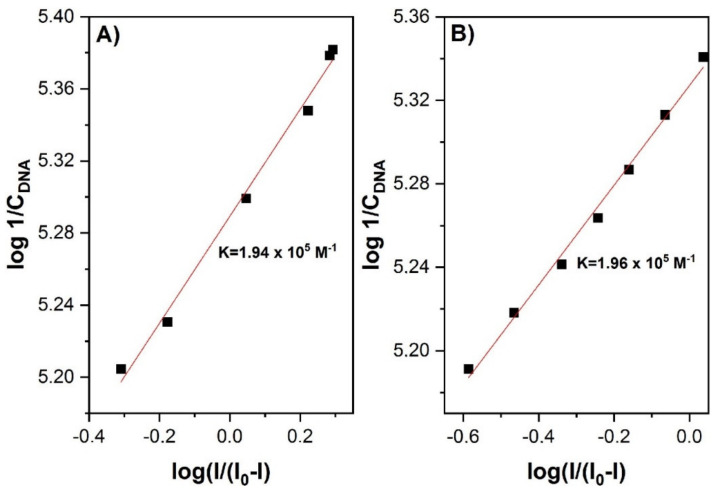
(**A**) The plot of log [1-(I_0_/I)] versus log [1/DNA] used to calculate the binding constant of 1,5-ppz-AQ-ctDNA complex; (**B**) The plot of log [1-(I_0_/I)] versus log [1/DNA] used to calculate the binding constant of the ethidium bromide–ct-DNA complex. Figure adapted from the reference [24] with permission from Elsevier.

**Figure 4 molecules-26-03478-f004:**
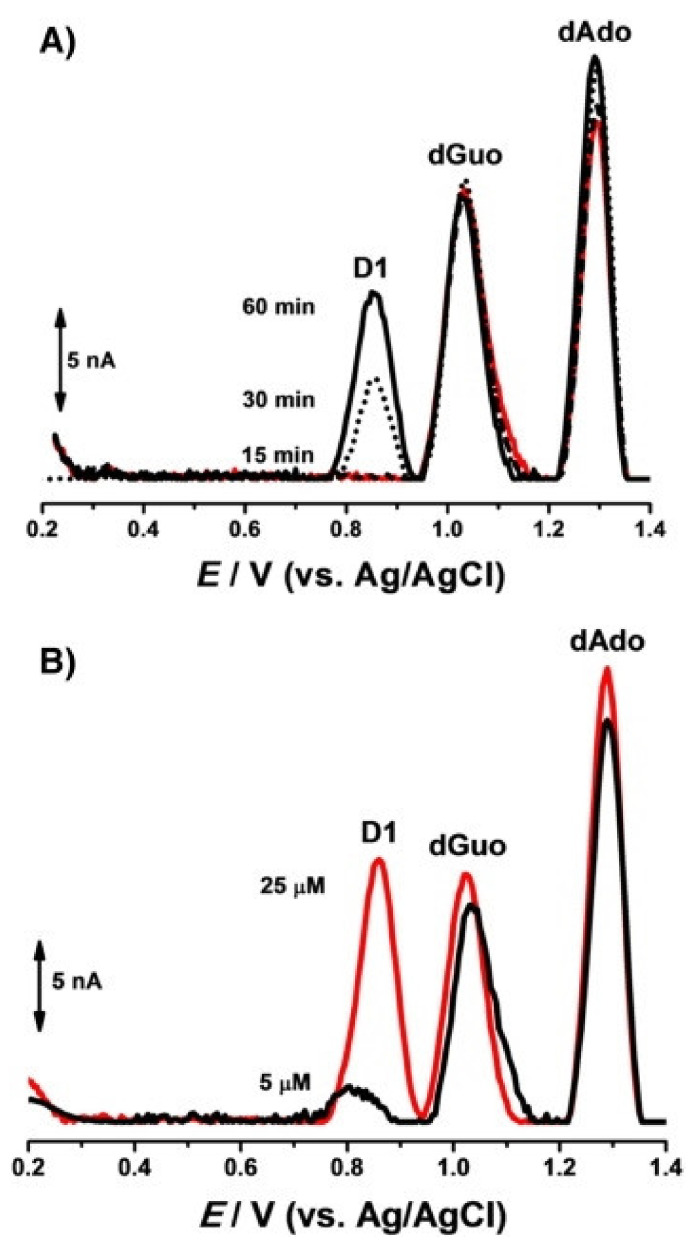
DP voltammograms, with no conditioning potential, in a 0.1 M acetate buffer with a pH of 4.5, with the dsDNA-electrochemical biosensor after: (**A**) (▬) 30 min in the buffer control experiment, and incubation in: 10 μM of danusertib during (•••) 15, (⁃⁃⁃) 30 and (▬) 60 min, and (**B**) incubation in (▬) 5 μM and (▬) 25 μM of danusertib during 30 min. Figure adapted from the reference [45] with permission from Elsevier.

**Figure 5 molecules-26-03478-f005:**
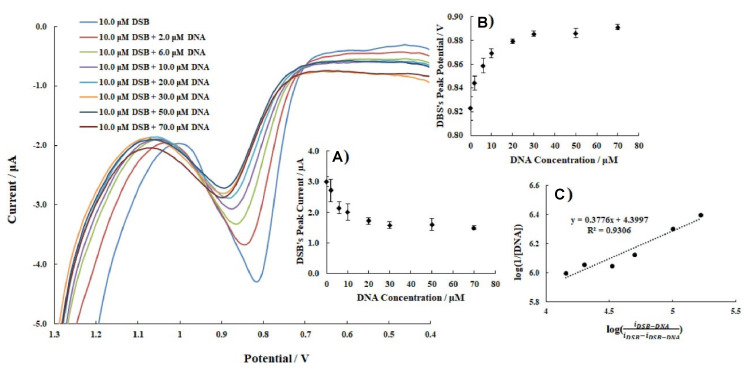
DPV voltammograms of 10.0 μM DSB with increasing concentrations of DNA in an acetate buffer solution with a pH of 4.8. Insets: (**A**) C_DNA_−i*_p_*_DSB_; (**B**) C_DNA_−E*_p_*_DSB_; (**C**) $\log\frac{i_{drug-DNA}}{i_{drug}-i_{drug-DNA}}-\log\left(\frac{1}{\left[DNA\right]}\right)$. Figure adapted from the reference [54] with permission from Elsevier.

**Figure 6 molecules-26-03478-f006:**
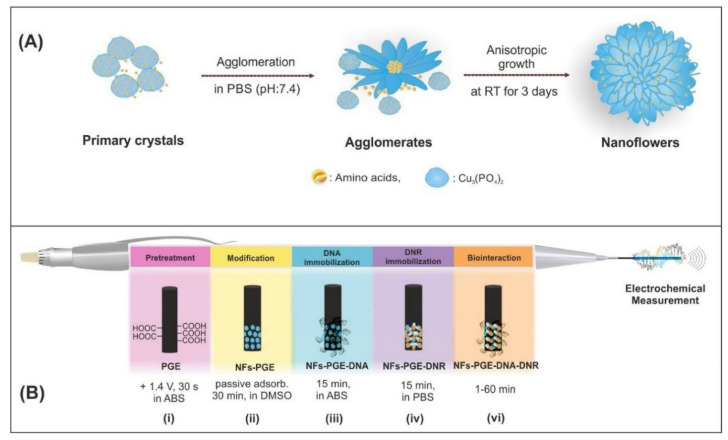
(**A**) Schematic illustration of the formation of amino acids-Cu_3_(PO_4_)_2_ hybrid NFs, (**B**) The representative scheme of the pretreatment of PGE (i), modification of NFs (ii), immobilization of DNA (iii) and DNR (iv), surface-confined interaction of DNR and ctdsDNA (vi). Figure adapted from the reference [53] with permission from Elsevier.

**Figure 7 molecules-26-03478-f007:**
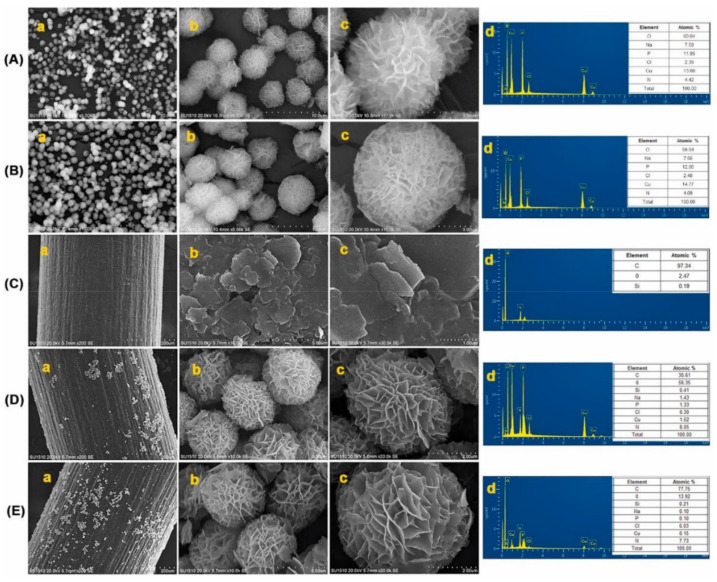
SEM image of (**A**) GNFs; (**B**) LNFs; (**C**) PGE; (**D**) GNFs-PGE; (**E**) LNFs-PGE ((**a**–**c**)—different resolutions) and EDX pattern (**d**). Figure adapted from the reference [52] with permission from Elsevier.

**Figure 8 molecules-26-03478-f008:**
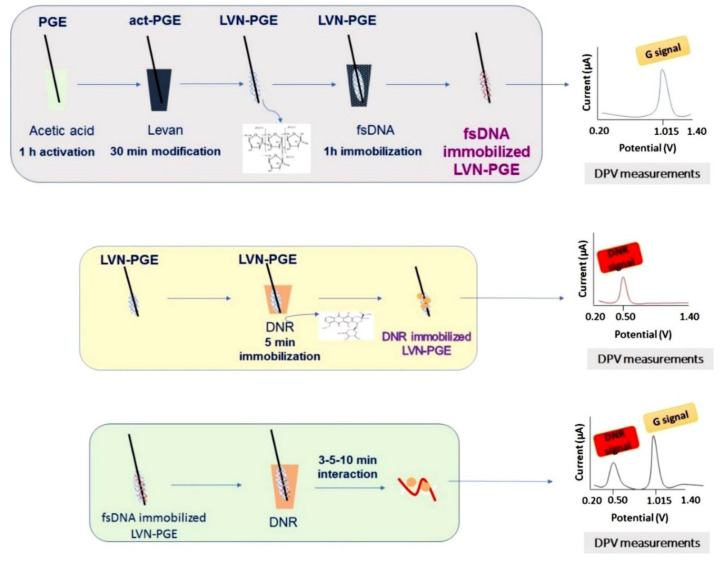
The experimental steps of the modification of LVN at the PGE surface, voltammetric determination of fsDNA and DNR using LVN-PGE and the voltammetric analysis of the biomolecular interaction between fsDNA and DNR at the LVN-PGE surface. Figure adapted from the reference [56] with permission from Elsevier.

**Figure 9 molecules-26-03478-f009:**
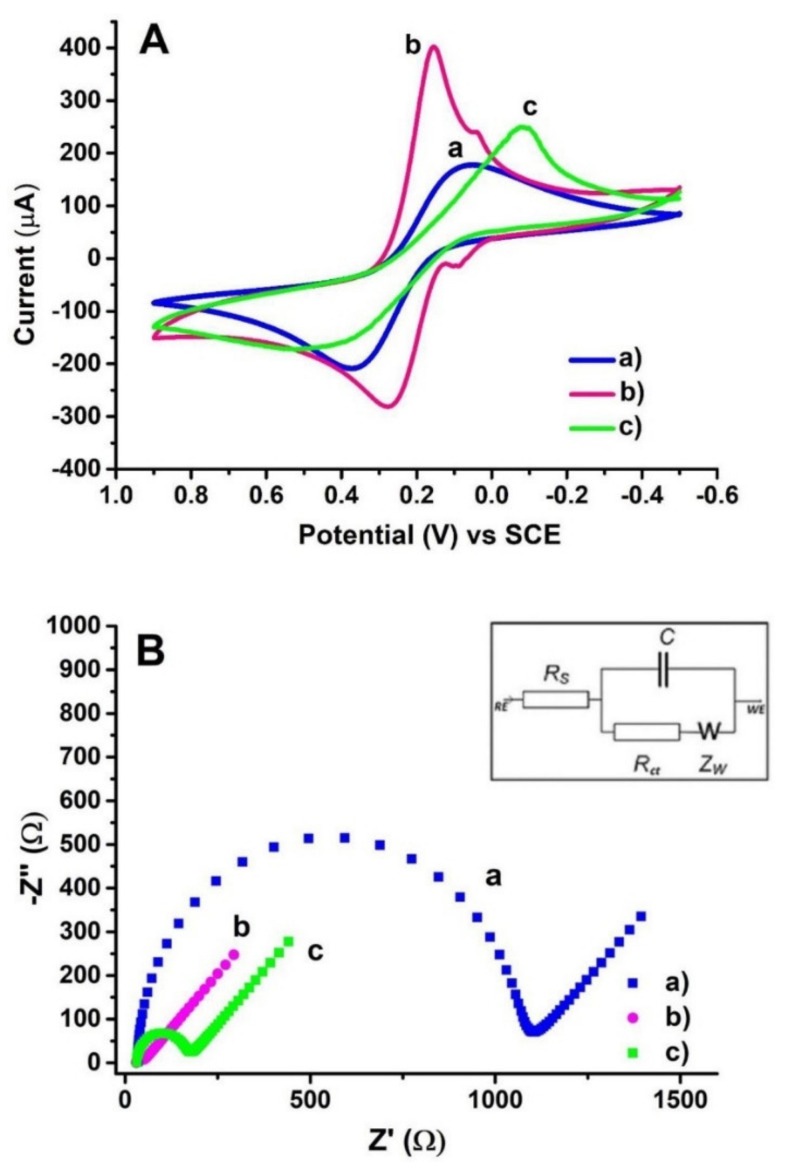
Electrochemical characterization of the surface layer assembly. (**A**) CVs at a scan rate of 100 mVs^−1^ and (**B**) The Nyquist diagrams of impedance recorded on (a) bare PGE, (b) PGE coated with poly(CTAB-MWCNTs), (c) 150 μg/mL dsDNA immobilized poly(CTAB-MWCNTs)/PGE in 0.1 M KCl solution containing 5.0 mM Fe(CN)_6_ ^3−^/^4−^ (Inset represents the equivalent circuit model for fitted impedance data. R_s_ is the solution resistance; R_ct_ is the charge transfer resistance at the electrode/electrolyte interface; C is the constant phase element related to the space charge capacitance at the electrode/electrolyte interface; W is the Warburg element). Figure adapted from the reference [55] with permission from Elsevier.

**Table 1 molecules-26-03478-t001:** Essential parameters of the described electrochemical research. Each column refers to: author, compound/drug studied, drug group, electrochemical method, type of electrode and medium used, respectively. Moreover, it contains the values of LOD/LOQ and the binding constant, as well as information about a suggested interaction mechanism.

Author [Ref.]	Compound/Drug	Drug Group	Method	Electrode	Medium	LOD/LOQ	Binding Constant	Interaction Mechanism
Satana et al. [41]	Clofarabine (CLF)	Anticancer	CV, DPV, SWV	GCE	pH 4.5 ABS	0.08 μM	NS	NS
Białobrzeska et al. [24]	1,5-di(piperazin-1-yl)anthracene-9,10-dione	Anticancer	CV	GCE	pH 7.4 PBS	NS	1.94 × 10^5^ M^−1^	Intercalation
Qin et al. [89]	ortho-5-fluorouracil meta-5-fluorouracil	Anticancer	CV	ctDNA/Au	pH 7.2 TBS	NS	2.33 × 10^3^ M^−1^ 1.45 × 10^3^ M^−1^	Groove
Bayraktepe [54]	Dasatinib (DSB)	Anticancer	CV, DPV	PGE	pH 4.8 ABS	NS	2.51 × 10^4^ M^−1^	Intercalation
Jabeen et al. [82]	morin (mor) quercetin (quer) primuletin (prim)	Potentially anticancer (Flavonoids)	CV	GCE	pH 7.4 PBS	NS	9.01 × 10^3^ M^−1^ 4.82 × 10^3^ M^−1^ 0.88 × 10^3^ M^−1^	Intercalation Groove Electrostatic
	Cu–morin (Cu–mor) Fe–morin (Fe–mor) Cu–quercetin (Cu–quer) Fe–quercetin (Fe–quer) Cu–primuletin (Cu–prim) Fe–primuletin (Fe–prim)	(Flavonoids complexes)	CV	GCE	pH 7.4 PBS	NS	12.0 × 10^3^ M^−1^ 0.53 × 10^3^ M^−1^ 0.92 × 10^3^ M^−1^ 0.89 × 10^3^ M^−1^ 18.02 × 10^3^ M^−1^ 9.89 × 10^3^ M^−1^	Intercalation Electrostatic Electrostatic Electrostatic Intercalation Intercalation
Buoro et al. [43]	Gemcitabine (GEM)	Anticancer	DPV	GCE,	pH 4.5 ABS	NS	NS	NS
				dsDNA/GCE				
Diculescu et al. [45]	Danusertib	Anticancer	DPV	GCE, dsDNA/GCE	pH 4.5 ABS	NS	NS	Electrostatic and forms a complex
Dindar et al. [47]	Citalopram (CIT) S-enantiomer—escitalopram (ESC)	Antidepressant	DPV	GCE	pH 4.7 ABS	NS	5.6 × 10^4^ M^−1^ 8.5 × 10^4^ M^−1^	Groove or electrostatic
Ponkarpagam et al. [49]	Rosiglitazone (RG)	Antidiabetic	CV, DPV	GCE	pH 7.3 TBS	NS	3.4 × 10^3^ M^−1^	Groove
Eksin et al. [51]	Daunorubicin (DNR)	Anticancer	EIS, DPV	cQD-PGE	pH 4.8 ABS	0.02 μg/mL	NS	NS
Findik et al. [53]	Daunorubicin (DNR)	Anticancer	DPV	NFs-PGE	pH 4.8 ABS	2.93 µM	NS	NS
Findik et al. [52]	Mitomycin C (MC)	Anticancer	DPV	NFs-PGE	pH 4.8 ABS	12.55 μg/mL	NS	NS
Bolat [55]	Irinotecan (CPT-11)	Anticancer	DPV	poly(CTAB-MWCNTs)/PGE	pH 4.8 ABS	1.03 μg/mL	6.84 × 10^4^ M^−1^	Groove
Janiszek et al. [48]	IPBD Cl-IPBD	Anticancer	DPV, ACV	scpUC19/GCE	pH 4.7 ABS	NS	NS	NS
Congur et al. [56]	Daunorubicin (DNR)	Anticancer	DPV	LVN-PGE	pH 4.8 ABS	510 nM	NS	Intercalation
Javar et al. [58]	Amsacrine	Anticancer	DPV	Eu^3+^-doped NiO/CPEs	pH 7.0 PBS	0.05 μM	NS	Intercalation
Kumar et al. [60]	[MnC_42_H_32_N_4_Cl_2_] [FeC_42_H_32_N_4_Cl_2_] Cl[CoC_42_H_32_N_4_Cl_2_] [NiC_42_H_32_N_4_Cl_2_]	Antibacterial	CV, DPV	Pt	pH 7.2 TBS	NS	3.19 × 10^2^ M^−1^ NS 4.23 × 10^2^ M^−1^ 3.69 × 10^2^ M^−1^	Intercalation
Temerk et al. [59]	Flutamide (Flu)	Anticancer	CV, SWV	HMDE	pH 7.4 PBS	NS	1.70 × 10^5^ M^−1^	Intercalation
Tajik et al. [90]	Taxol	Anticancer	DPV	dsDNA/PGE	pH 4.8 ABS	8.0 × 10^−8^ M	NS	Intercalation

Abbreviations included in the table: NS—not stated; CV—cyclic voltammetry; DPV—differential pulse voltammery; SWV—square wave voltammetry; GCE—glassy carbon electrode; PGE—pencil graphite electrode; CPE—carbon paste electrode; EIS—electrochemical impedance spectroscopy; ACV—alternating current voltammetry; Pt—platinum electrode; HMDE—hanging mercury drop electrode; ABS—acetate buffer solution; PBS—phosphate buffer solution; TBS—tris buffer solution.

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
