# Peer review of "What Can Electrochemical Methods Offer in Determining DNA–Drug Interactions?"

_molecules, 2021, doi:10.3390/molecules26113478_

Round 1

Reviewer 1 Report

I carefully read the revised version of the manuscript and noticed a significant improvement in the quality of the study. In this form, I can recommend this study for publication in Molecules.

Author Response

Thank you very much for your acceptance.

Reviewer 2 Report

I am sorry but I am not satisfied with the modifications,  variations carried out. The review is still too long, and English need more deep revision. Paragraph 1.2 is completely unuseful: it is unacceptable to see that SPR is an absorption method

Author Response

“I am sorry but I am not satisfied with the modifications, variations carried out.”

Satisfaction is an individual case. We do not think that our review paper of type is still too long. Now it consists of 49 pages along with figures on separate page each, table, and references. This paper, if accepted, would cover about 10-11 pages of a regular issue, what is not too much for a review-type.

“The review is still too long, and English need more deep revision.”

I can only agree, as foreigners we are not well-skilled in English. But I also have an impression that the Reviewer should not behave as a superior to me or others. The Reviewer wrote in hers/his present review version: “(...)and English need (...).. In this part of the sentence I also found a huge grammar mistake, usually avoided by beginners. It should be: ““(...)and English needs (...).” Our final version of the paper was checked and corrected by a native speaker. We did our best in this case.

“Paragraph 1.2 is completely unuseful: it is unacceptable to see that SPR is an absorption method”

This sentence of the Reviewer’s comments is also confusing. Previously, two out of three Reviewers asked for comparison of difference between analytical techniques related to DNA-drug: advantages of electrochemical methods with respect to the other. So, we did it in the revised version of the paper. Now, the Reviewer has changed hers/his mind and calls it unuseful. We decided not to remove that part of our manuscript. We have also never statet and written in any version of our manuscript that the SPR was an absorption method. Moreover, because we have an impression that the Reviewer read the paper not very carefully [one of hers/his previous comments was also not true: “How primary structure can change upon intercalation???” Our answer was: We have never written such statement in any version of our manuscript. It is obvious that primary structure does not change upon intercalation. In the subchapter 1.1. it is clearly stated that After intercalation process, the primary and secondary DNA structures remain unchanged.” (page 3, line 4 from the bottom)] we have extended the paragraph about the SPR method (please see last ten lines of page 5 of the present manuscript): “SPR is an optical technique that allows to determine the concentration of biomolecules at a short distance from a metal surface by measuring changes in light refraction parameters. The basis of the SPR is the interaction of an incoming light source with a thin metallic film in close contact with a prism or grating. SPR is the basis of many standard tools used for measuring adsorption of material onto planar metal (typically gold or silver) or metal nanoparticle surfaces. In order to detect an interaction, one molecule is immobilized onto the sensor surface and its binding partner is injected in sample buffer [37–39]. The main advantages of SPR over other methods is that no labeling is required, the amount of both ligand and analyte needed to obtain satisfying results is low and the experiment is relatively rapid. The limitation of this technique on the other hand, is that it cannot verify the stability of the complex formed during drug binding to DNA [34].”

Round 2

Reviewer 2 Report

I am completely unsatisfied with the revised paper. SPR section is completely far from the focus the review and it appears still too long and verbous

Author Response

I have an impression that the lack of satisfaction has become rather personal than substantive. Anyway, the SPR was described in a much more condensed way. That subchapter has been shortened. (please see the highlighted text of page 5 of the revised manuscript)

This manuscript is a resubmission of an earlier submission. The following is a list of the peer review reports and author responses from that submission.

Round 1

Reviewer 1 Report

It is important to design DNA-targeting drugs. Electrochemical approaches enable to determine the type of interactions, binding strength, and DNA damage caused by DNA-targeting molecules. In this manuscript, the authors summarized the electrochemical approaches for studying the interactions of small molecules with double stranded DNAs. The review may help research community to understand recent development on the electrochemical approaches and methods used to study the interactions between pharmaceuticals and DNA. The manuscript is well writing, which could be published as it.

Reviewer 2 Report

The review manuscript sent by Makowski and collaborators is of interest to the readers of the journal Molecules, dealing with a current topic, namely the evaluation of the interaction between drugs and DNA for understanding the mechanisms of therapeutic action.

The manuscript is written quite well, but there are some issues that need to be addressed, for which the authors should take into account the following observations and suggestions:

A more detailed, comparative presentation of the methods used to describe DNA-molecule interactions should be made by the authors in subchapter 1.3. Their simple enumeration does not allow the reader to get a complete idea of their advantages and disadvantages for the mentioned application.

For the part of the examples of approaches based on electrochemical methods for the study of the interaction of DNA with drug substances is a descriptive presentation. Here should be introduced some critical comparative discussions and a table showing the important parameters for this type of study. This would make the manuscript easier for readers to read and understand.

Some figures are repetitive and can be removed (eg.: Figure 2 and Figure 5), or combined figures can be made in which to insert portions of similar figures from several articles for comparative presentation.

The quality of all the figures is very low, this should be solved.

Before concluding, a section on discussions and future trends should be introduced, whereby the authors discuss the information presented and establish current trends in the field.

The conclusions section should also be developed because it seems to me to be telegraphic and seems to repeat the information in the abstract.

Reviewer 3 Report

The review entitled “What can electrochemical methods offer in determining DNA-drug interactions?” focuses on electrochemical methods as in vitro binding assay for the detection and quantification of DNA-drug interaction

Even it is very ambitious, it suffers of an excessive length and many English mistakes that make it very hard to read. 

I advise the authors to shorten/modify several parts. First of all, the abstract is completely unclear: it is written very badly and no focus neither the advantages of the review are outlined. In addition, the Introduction is too generic with very poor scientific sentences many mistakes... intercalare is not latin but Italian, intercalating compounds not necessary are positioned perpendicularly to the axis of the double helix. How primary structure can change upon intercalation??? I advise changing it to compare differences among analytical techniques related to DNA-drug: advantages of electrochemical methods with respect to the other.

All paragraphs are too generic and case studies are reported as a shopping list instead of probing examples. the electrochemical sensing probe should be mentioned as well as the scientific information deriving from this approach in the quantification of interaction